# Intracellular ATP Concentration and Implication for Cellular Evolution

**DOI:** 10.3390/biology10111166

**Published:** 2021-11-12

**Authors:** Jack V. Greiner, Thomas Glonek

**Affiliations:** 1The Schepens Eye Research Institute of Massachusetts Eye & Ear Infirmary, Boston, MA 02114, USA; 2Department of Ophthalmology, Harvard Medical School, Boston, MA 02114, USA; 3Clinical Eye Research of Boston, Boston, MA 02114, USA; tglonek@rcn.com

**Keywords:** adenosine triphosphate, ATP, hydrotropic function, interspecies, protein aggregation

## Abstract

**Simple Summary:**

The lens organ of the eye and muscle tissue exist at opposite ends of the metabolic spectrum. Lens is metabolically quiescent using little energy, while muscle has high energy requirements. However, both tissues contain excessively high millimolar concentrations of adenosine triphosphate (ATP) molecules, the biochemical energy source for all of life’s activities. Nature does not manufacture excess ATP molecules, and thus the question becomes: Why this dichotomy? To answer this question, we prepared a compilation of measured ATP concentrations from cells/tissues/organs across three phylogenetic domains of life: eukaryotes, archaea, and prokaryotes. Among a total of 136 organ/tissue/cell sources, we found that all specimens examined contained excessively high concentrations of ATP (average~4.4 millimolar), regardless of how they lived or their function. Since these specimens only require a small micromolar amount of ATP for energy metabolism, this observation reinforced the notion that ATP has another major function in life. The recent demonstration that ATP acts as a protein solvating agent keeping proteins in their active forms prevents aggregation and in high concentrations may avert a multitude of diseases. The presence of high concentrations of ATP across phylogenetic domains suggests another role for ATP fundamental to cellular/tissue/organ function and biological, biochemical, and biophysical evolution.

**Abstract:**

Crystalline lens and striated muscle exist at opposite ends of the metabolic spectrum. Lens is a metabolically quiescent tissue, whereas striated muscle is a mechanically dynamic tissue with high-energy requirements, yet both tissues contain millimolar levels of ATP (>2.3 mM), far exceeding their underlying metabolic needs. We explored intracellular concentrations of ATP across multiple cells, tissues, species, and domains to provide context for interpreting lens/striated muscle data. Our database revealed that high intracellular ATP concentrations are ubiquitous across diverse life forms including species existing from the Precambrian Era, suggesting an ancient highly conserved role for ATP, independent of its widely accepted view as primarily “metabolic currency”. Our findings reinforce suggestions that the primordial function of ATP was non-metabolic in nature, serving instead to prevent protein aggregation.

## 1. Introduction

Crystalline lens and striated muscle are at opposite ends of the metabolic spectrum. Lens is a metabolically quiescent tissue, whereas striated muscle is a mechanically dynamic tissue with high-energy requirements, yet both tissues contain millimolar levels of adenosine triphosphate (ATP) ([>2.3 mM] [1,2,3,4]), far exceeding their underlying metabolic needs. In the case of the lens [1], this millimolar concentration of ATP was three orders of magnitude greater than what is believed to be required for known functions of ATP (metabolic and otherwise) (Table 1), raising the possibility of an additional heretofore unrecognized and unappreciated biochemical role for ATP.

Recently, Patel et al. [5] presented compelling evidence that ATP functions as a hydrotrope when present in millimolar concentrations in extracts of cellular and tissue homogenates, preventing protein aggregation and maintaining protein solubility. This proposed mechanism involves both the negatively charged hydrophilic tripolyphosphate moiety and the relatively hydrophobic adenine ring, conferring amphiphilic properties to the ATP molecule that allow it to act essentially as a surfactant. These surfactant properties allow ATP to bind to the hydrophobic surfaces of proteins, thus encouraging intracellular proteins to maintain a fluid state sufficient to prevent nonfunctional protein aggregation. Preventing nonfunctional protein aggregation allows intracellular proteins to maintain their inherent biological, physiological, and homeostatic activity, all of which impact cellular and organismal survival. In light of this, we addressed the possibility of a hydrotropic function of ATP in the crystalline lens and found it to be conceptually valid at a biophysicochemical level [6]. The question was then raised as to whether this finding of millimolar intracellular ATP in the crystalline lens (and its apparent ability to function as a hydrotrope preventing protein aggregation) was an anomaly or a phenomenon that might be more widespread in nature. As such, the purpose of this study was to explore intra- and interspecies ATP concentrations in normal cells, tissues, and organs to determine whether a millimolar concentration of ATP is a common feature across species and domains and to provide an evolutionary context for these observations.

## 2. Materials and Methods

Using a literature search focused on the concepts “adenosine triphosphate”, “intracellular”, and “ex vivo”, 442 published studies in Medline via PubMed (1946–2020) articles were retrieved. Articles were examined for the concentration of ATP in normal cells, tissues, and organs under study. Although populating and establishing the database was designed to be comprehensive, in order to avoid the possible influence of cellular abnormalities on the data, reports on the concentration of ATP in laboratory cell lines were excluded. This exclusion was required, since ATP’s role in the natural evolution of normal cellular biochemistry may not be maintained and be representative of normal cellular ATP in that such laboratory cell lines ordinarily are evolved from abnormal cell lines. Measurements of ATP in the cellular cytoplasm and various individual intracellular organelles also were excluded. Further, studies reporting tissue ATP concentrations in the micromolar range also were excluded, as these extremely low concentrations could not be rationalized with the bulk of the captured data, particularly in those cases where reports from other laboratories on the same tissue yielded ATP concentrations in the millimolar range; for example, ATP molarity was measured in brain in two comparable reports [7,8], both using the luciferin–luciferase assay for ATP molarity. The first study determined the ATP molarity to be 5.9 mM, whereas the second study obtained 121 µM. These values differ by over two orders of magnitude, a finding that raises issues with the luciferin–luciferase assay as applied to biological specimens. It appears that strong acid extraction of tissue ATP ordinarily is required to obtain the highest tissue ATP molarities, and such findings have been validated using 31P NMR spectroscopy of intact living tissues [1,2,3].

One hundred thirty-six (Appendix A) of the retrieved studies had ATP concentrations measured in millimolar values or provided data that permitted mathematical conversion to conform to millimolar units, allowing comparison between studies. All studies where the concentration of ATP was reported or converted to millimolar units were included in the data sets with the exception of 9 outliers (>3 standard deviations from group means). No filters for language, study design, or date of publication were used in the literature search. References cited in all relevant articles were examined to identify and include additional published studies. Only ATP concentrations in cells/tissues/organs presumed normal were tabulated.

One hundred twenty-seven statistical cases were organized into 7 groups: Cardiac Muscle (N = 15), Skeletal Muscle (46), Brain (8), Liver (16), Retina (5), Other (a compilation of other cells, tissues, and organs) (16), and Microorganisms (21). “Other” refers to cells, tissues, and organ sample studies that did not achieve a significant enough number of cases (>3) to be considered as their own group. The data set, means, and standard deviations are presented; t-test comparisons among all groups without outliers were computed and tabulated.

## 3. Results

The average concentration of ATP in all normal whole source cells and tissues presented in Table 2 (outliers excepted) was 4.41 mM. Normal cell and tissue groups, though heterogeneous, all exhibited similar high millimolar concentrations of ATP (Table 2). Most intracellular ATP values were obtained from established laboratory animals and microorganisms (Appendix A) [7,9,10,11,12,13,14,15,16,17,18,19,20,21,22,23,24,25,26,27,28,29,30,31,32,33,34,35,36,37,38,39,40,41,42,43,44,45,46,47,48,49,50,51,52,53,54,55,56,57,58,59,60,61,62,63,64,65,66,67,68,69,70,71,72,73,74,75,76,77,78,79,80,81,82,83,84,85,86,87,88,89,90,91,92]. The greatest number of compiled studies was on muscle tissues, although there were a substantial number of studies on brain and liver as well as microorganisms, along with a variety of other tissues (Other group).

The most common method employed for determining ATP concentrations in all cells, tissues, and organs studied used denaturants/extracting solvents containing substantial quantities of strong acids (91 cases, 72%), the most common of these being perchloric acid, usually about the 6% level (Appendix A). The four most common quantitative analytical technologies employed (Appendix A) were NADH/TPNH enzymatic ATP-coupled reactions monitored through spectrophotometry (35 studies, 27%), coupled luciferin–luciferase spectral-fluorescence (32 studies, 25%), HPLC (27 studies, 21%), and NMR (26 studies, 20%). Other methods were either less frequent or not stated in the source study. When the method of assay for ATP was not stated in the published reports, even by searching previous method references, these ATP values were excluded from this compilation of reports. Moreover, the consistency of the data, including the methods used in Appendix A, demonstrates the similarity in the ATP concentrations and that regardless of the laboratory conducting the experiments or the analytical methods employed, the concentration of ATP is a constant in living cells/tissues/organs and whole organisms.

Of particular note is that the mean and standard deviation of the Other group were essentially the same as that of the Brain, Liver, and Microorganism groups (Table 2 and Appendix A). Of the cells and tissues compiled within the Other group, none could be reasoned to require a concentration of ATP significantly above basic metabolic needs. Among the reports studied, the average concentrations of ATP in the Cardiac Muscle and Striated Muscle groups were more than twice the concentration of ATP in the Brain, Liver, Other, and Microorganism groups (Table 2). Moreover, the average concentration of ATP in cardiac muscle exceeded skeletal muscle by >21%.

Considering pairwise comparisons among the Brain, Liver, Retina, Other, and Microorganism groups (Table 3), there were no statistically significant differences (*p* > 0.95), except for Retina vs. Other, and the means among these groups averaged to 2.71 mM (Table 2). In contrast, comparisons of either the Striated or Cardiac Muscle groups with Brain, Liver, Other, and Microorganism groups showed these to be significantly different. Nevertheless, the mean concentrations of ATP of all seven groups were in the millimolar range.

The compiled groups not only include anatomically diverse animal kingdom cells, tissues, and organs but also include species representing different taxonomic domains including eukaryotes, archaea, and prokaryotes. The data do show that cardiac and skeletal muscle groups are different from the other four groups except for Retina (Table 3); nevertheless, all group concentration levels are clustered within the range of 1.92–7.47 mM.

## 4. Discussion

This meta-analysis demonstrates that millimolar intracellular concentrations of ATP (averaging 4.41 mM) are a common ubiquitous feature across a diverse spectrum of life forms, including cells from prokaryotic, archaeotic, and eukaryotic domains (Appendix A). This common ATP high-concentration feature was unexpected, since the groups of cells and tissues studied were clearly not similar. Moreover, the concentration was orders of magnitude above the micromolar levels of ATP necessary to conduct the principle known role of ATP as the energy source for metabolism as well as to carry out the established functions of intracellular ATP (Table 1).

The range of high intracellular ATP concentrations reported herein is consistent with the previously reported physiological concentration of ATP ranging from 2–8 mM [5,93]. The range of variations among laboratories in ATP determinations (Appendix A) must include consideration of the quality, sophistication, and precision of the instrumentation used to make such determinations as well as the training and level of expertise of investigators. Examination of the Appendix A data indicated that a three-standard-deviation range was an appropriate inclusion criterion. Further, the closeness of the standard deviations observed among the groups studied (Table 2) provides a statistical indication for the level of precision and accuracy in the reported analyses.

To a first approximation, the higher concentration of ATP among the two muscle groups can be attributed to the fact that muscles are mechanical contractile tissues that consume a large amount of energy upon sudden demand. It follows, therefore, that the internal ATP reservoir in these types of muscles would be larger than that of the other five groups studied. With the exception of the retina, the four other tissue samples included in this study would be expected to maintain metabolism at essentially a constant rate and, unlike muscle tissue, would not experience large sudden energy demands requiring an ATP reservoir, yet, all of the tissues examined, regardless of the demands they may meet, maintain intracellular ATP concentrations within a fairly narrow high millimolar range. The ATP concentration necessary for the energy demand of cells and tissues ordinarily only requires micromolar levels. Nevertheless, the millimolar concentration of ATP in all these different tissue groups, including microorganisms, coupled with the known phenomenon of the conservation of cellular and tissue ATP, indicates that the high level of ATP must be maintained for some fundamental function other than as an energy reserve.

The amphiphilic property of ATP permits it to function as a hydrotrope [5,94] by binding to the hydrophobic surfaces of proteins [5]. Patel et al. [5] demonstrated that ATP in high millimolar concentrations was much more efficient than classical hydrotropes for creating a microenvironment for the prevention of aggregation of hydrophobic protein molecules. Patel et al. [5] demonstrated this in experiments using cell and tissue homogenates. We showed supporting evidence for such hydrotropic interactions of ATP in a whole intact functioning organ, the crystalline lens [6], using 31P NMR spectroscopy. Protein aggregation in the lens organ with cells containing the highest concentration of protein in the body has been shown to result in opacification that results in loss of transparency (cataractogenesis [6] and, thus, lens focusing function (presbyopiogenesis [95]). Evidence of the hydrotropic interaction with ATP [5,6] supported our hypothesis that ATP at high intralenticular concentrations can prevent protein aggregation by maintaining intracellular proteins in a fluid state permitting maintenance of organ functional transparency. Maintaining high concentration of ATP in lenticular tissue with such a low metabolic activity coupled with the findings of Patel et al. [5] supports the premise that ATP prevents protein aggregation and can maintain the lens proteins’ inherent biological activity.

In accord with the more highly specialized cellular tissues cited herein, single-cell microorganisms similarly have high concentrations of ATP (Table 2 and Appendix A). Moreover, since some of the organisms in the Microorganism group are known to have ancient roots with a heritage extending at least 2.5 billion years [87,96,97], evolutionary heredity may be considered. Since high concentrations of ATP are present in contemporary relatives of these organisms, known to have ancestors existing in the Precambrian Era when primordial proteins were evolving, it is suggested that ATP’s initial function may have been as a hydrotrope, predating ATP’s use as intracellular energy currency. Hypothetically, bacteria with ancient genomic roots that maintain function without a nucleus, mitochondria, or other intracellular organelles that produce and/or utilize ATP at high concentrations indicate that the millimolarity of ATP observed in contemporary organisms may have been a feature of the first functional progenotes.

Protein aggregation has been shown to result in cellular dysfunction and eventual death of cells, tissues, and organisms because such aggregation changes protein shape and the nature of available protein surfaces or sites of activity, inhibiting or preventing function. Although protein aggregation can occur when proteins are too closely approximated, proteins in order to function need to be close enough that they interact, but not so close that they result in pathological protein aggregation. Considering the foregoing and hypotheses proposed [5,6], if ATP is in part or wholly responsible for maintaining an appropriate distance between proteins to allow proper function and avoid protein aggregation, the findings presented herein imply that there exists a critical concentration for intracellular ATP, which from Table 2 appears to be just under 3 mM.

This study has limitations. Since the meta-analysis of the literature citing measurement of the ATP concentrations spans the period from 1946 through 2020, the results presented were derived from methods of measuring the ATP concentration that evolved with more precise technologies. However, with evolution of these methods, it must be cautioned that later methods, for example, the luciferin–luciferase reaction, require consideration and calibration to demonstrate their accuracy. This is particularly so when more recent values obtained using such methods are lower by three orders of magnitude than those determined earlier. Such reports as described in the Methods were not included in this meta-analysis. Other limitations as explained in the Methods were the exclusion of reports on ATP concentrations utilizing modified cellular/tissue preparations: these include cell lines to avoid the possible influence of cellular abnormalities; use of cytoplasm and various intracellular organelles to avoid the effects of both mechanical and chemical disturbances induced as a result of sample procurement; reports with data in the micromolar range with exceedingly low concentrations of ATP inconsistent with the bulk of the captured data in the millimolar range (this phenomenon was often associated with conflicting reports employing the luciferin–luciferase reaction for measurement of ATP demonstrating very low ATP concentrations in the same tissue and even the same species). Although these selective limitations may result in some degree of bias, it is our contention that such limitations are justified. Although our interpretation of the data presented herein is albeit unidirectional, the basis that ATP is acting as a hydrotrope preventing protein aggregation in living systems is established by multiple experiments [5,6]. In the absence of any other explanation or hypotheses, we presented our hypothesis for the function of high concentrations of ATP in the intact crystalline lens and other cells/tissues.

## 5. Conclusions

Considering the findings herein of high millimolar concentrations of ATP throughout the phylogenic landscape (Appendix A), it appears that ATP acting as a hydrotrope may play a major functional role in preventing protein aggregation in diverse lifeforms. Taken collectively, the intracellular ATP concentration data presented herein demonstrate that high millimolar intracellular concentrations of ATP across taxonomic domains and kingdoms are a fundamental feature of a living cell. Moreover, this finding supports the hypothesis that millimolar concentrations of intracellular ATP serve a critical and foundational molecular function for organismal homeostasis. Considering the heritage of some of the microorganisms studied, this phenomenon may have been a feature of early progenotes when proteins were first being enclosed within a cell membrane and where the hydrotropic feature of ATP prevented nonproductive protein aggregation and maintained protein solubility.

## Figures and Tables

**Table 1 biology-10-01166-t001:** Functions of ATP.

A molecular carrier of intracellular energy
The ultimate metabolic source of high-energy phosphate bonds
The parent residue giving rise to vitamin dinucleotides and other cofactors
An allosteric enzyme regulator for modulating protein activities
The principal metabolite for cellular energy transduction mechanisms
The transport of macromolecules, such as proteins, into and out of cells
A phosphorylating agent in phosphate regulation of transmembrane proteins
A source of the adenosine nucleotide, one of the 4 letters of the genetic code
A hydrotropic functional molecule preventing intracellular protein aggregation [5]

**Table 2 biology-10-01166-t002:** ATP concentrations in normal cells/tissues/organs computed from Appendix A (means ± SD).

Cell/Tissue/Organ Group	mM Concentration (Mean ± SD)	N
Muscle (cardiac)	7.47 ± 4.12	15
Muscle (skeletal)	5.86 ± 1.91	46
Brain	2.88 ± 1.33	8
Liver	2.92 ± 1.98	16
Retina	4.14 ± 2.06	5
Other	1.92 ± 0.95	16
Microorganisms	2.74 ± 2.55	21
Non-muscle Group	2.71 ± 1.97	66
All Groups	4.41 ± 2.93	127

SD, standard deviation. N, number of cases. Other group, compilation of cell/tissue/organ studies not >3 considered to be a separate group. Non-muscle Group, Brain, Liver, Retina, Other, and Microorganisms combined. All Groups, Cardiac and Skeletal Groups plus non-muscle groups.

**Table 3 biology-10-01166-t003:** Pairwise comparisons of ATP concentrations in normal cells/tissues/organs groups.

Groups Compared	Total N	t-Statistic	*p*-Value	Significantly Different *
Cardiac vs. Skeletal	61	2.0698	0.0429	X
Cardiac vs. Brain	23	3.0396	0.0062	X
Cardiac vs. Liver	31	3.9564	0.0005	X
Cardiac vs. Retina	20	1.7153	0.1035	
Cardiac vs. Other	31	5.2471	0.0001	X
Cardiac vs. Microorganisms	36	4.2569	0.0002	X
Skeletal vs. Brain	54	4.2239	<0.0001	X
Skeletal vs. Liver	62	5.2485	<0.0001	X
Skeletal vs. Retina	51	1.9027	0.0630	
Skeletal vs. Other	62	7.8852	<0.00001	X
Skeletal vs. Microorganisms	67	5.5739	<0.00001	X
Brain vs. Liver	24	0.0594	0.9531	
Brain vs. Retina	13	−1.1867	0.2500	
Brain vs. Other	24	2.0379	0.5376	
Brain vs. Microorganisms	29	0.1450	0.8664	
Liver vs. Retina	21	−1.1867	0.2500	
Liver vs. Other	32	1.8322	0.0769	
Liver vs. Microorganisms	37	0.2396	0.8120	
Retina vs. Other	21	3.4103	0.0029	X
Retina vs. Microorganisms	26	1.1367	0.2669	
Other vs. Microorganisms	37	−1.2234	0.2293	
Non-muscle vs. Cardiac	81	−6.6829	<0.00001	X
Non-muscle vs. Skeletal	112	−8.4348	<0.00001	X
Non-muscle vs. Cardiac + Skeletal	127	−8.5444	<0.00001	X

N, number of cases. t-statistic, two-sample equal variance *t*-test. * significance <0.5. Other, combination of cells/tissues/organs not >3 considered to be a separate group. Non-muscle cells/tissues/organs, Brain, Liver, Retina, Other, and Microorganisms combined.

## Data Availability

Data supporting reported results are provided in the references cited in Appendix A.

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
