# Peer review of "Intracellular ATP Concentration and Implication for Cellular Evolution"

_biology, 2021, doi:10.3390/biology10111166_

Round 1
Reviewer 1 Report
This is an excellent short article addessing a topic of increasing relevance. I have no substantial criticisms.
I read with interest the MS and felt that it was convincing and very well written. Often Reviewers struggle to make comments on a MS even if it is not necessary simply because "you have to", otherwise the Editor will get the impression that you did not do a proper job.
I firmly believe that criticisms are needed only when they are needed, and it's useless to impose an unneeded burden on the Authors.
Referees must be able to plainly say that a MS is good as it is, as on the other hand they must be ready to raise all due criticisms and plainly say that a MS is not suitable for publication.
Author Response
This is an excellent short article addessing a topic of increasing relevance. I have no substantial criticisms.
I read with interest the MS and felt that it was convincing and very well written. Often Reviewers struggle to make comments on a MS even if it is not necessary simply because "you have to", otherwise the Editor will get the impression that you did not do a proper job.
I firmly believe that criticisms are needed only when they are needed, and it's useless to impose an unneeded burden on the Authors.
Referees must be able to plainly say that a MS is good as it is, as on the other hand they must be ready to raise all due criticisms and plainly say that a MS is not suitable for publication.
We thank reviewer 1 for reviewing our manuscript.
Reviewer 2 Report
The authors performed a meta-analysis of literature on cellular concentrations of ATP to provide evidence for a recently described novel role of ATP as a hydrotrope. Ultimately analysing in excess of 120 publications the authors find that cellular ATP concentrations are in the millimolar range in all cells and across a diverse spectrum of life forms, including cells from prokaryotic, archaeotic, and eukaryotic domains. Based on this finding the authors conclude that high millimolar intracellular concentrations of ATP is a fundamental feature of a living cell and supports the hypothesis that these serve a critical and foundational molecular function for organismal homeostasis.
Although potentially interesting and valuable to have a summarized overview of ATP concentrations across such a wide spectrum of tissues, specimens and even taxa there are also some substantial shortcomings of the data collection and interpretation of the data.
Also given the style and content of the manuscript this is rather a review-like manuscript, providing a good overview of current studies, but limited new findings. I therefor suggest to redact the manuscript into a review format and include also reports that do not unidirectionally support the outline hypothesis.
Major points:
- Overall, the study is based on a literature search with heavy subsequent filtering of hits. This filtering does not seem to be appropriate in all cases and in some places seem to be biased.
E.g. why do the authors exclude “measurements of ATP in the cellular cytoplasm”. Selectively analyzing such measurements would be much more appropriate to substantiate the hypothesis of ATP serving a role as hydrotrope, rather bulk measurements of cellular ATP which “average” ATP concentrations across multiple intracellular organelles with highly divergent ATP levels and hence depend on organelle volume/density etc..
The authors also exclude studies reporting tissue ATP concentrations in the micromolar range based on the rationale that these are not in line with the bulk of other captured data. This is not a valid reason for exclusion as long as the technical approach for measuring ATP concentrations in these studies is sound. Even more so as the authors include studies in which the quantitative analytical technologies for measuring ATP were “not stated in the source study”.
I rather suggest the authors provide a full list of reported ATP concentrations (in non cell line specimens) and then discuss reasons for different outcomes rather than excluding the majority of studies a priori.
- The main concern is the rather unidirectional interpretation of the data to support the hydrotrope hypothesis. Although interesting, the fact that ATP concentrations are high across all examined specimens is not a proof for (an evolutionary) hydrotrope function of ATP. If the authors want to support and substantiate such claims, they need to compare ATP concentrations in specific intracellular compartments (e.g. cytoplasm) between healthy specimens and diseased specimens in which disease is caused by protein aggregation/misfolding.
- The statistical comparison of different specimens and Table 3 are not really informative and do not add to the paper.
Minor points:
- Table 1 lists functions of ATP which the authors claim require only micromolar concentrations of ATP. Such concentrations, and also references supporting these findings, should be included in the table. Otherwise, it’s a bit like a textbook list.
- The simple summary, contains a few grammatical errors. E.g.:
Line 7: the comma after prokaryotes should be replaced by a full stop.
Line 8/9: “(average~4.4 millimolar). regardless of how it lived or its function.” Should be changed to “(average~4.4 millimolar), regardless of how they lived or their function.”
- Some of the datasets in table S1 appear redundant. E.g. multiple data extracted from refs 41 or 67 are identical. Were these ATP concentrations clearly obtained from individual samples or were same samples mentioned several times in the respective references
Author Response
Comment #1. The authors performed a meta-analysis of literature on cellular concentrations of ATP to provide evidence for a recently described novel role of ATP as a hydrotrope. Ultimately analysing in excess of 120 publications the authors find that cellular ATP concentrations are in the millimolar range in all cells and across a diverse spectrum of life forms, including cells from prokaryotic, archaeotic, and eukaryotic domains. Based on this finding the authors conclude that high millimolar intracellular concentrations of ATP is a fundamental feature of a living cell and supports the hypothesis that these serve a critical and foundational molecular function for organismal homeostasis.
Although potentially interesting and valuable to have a summarized overview of ATP concentrations across such a wide spectrum of tissues, specimens and even taxa there are also some substantial shortcomings of the data collection and interpretation of the data.
Answer #1. We presume the substantial shortcomings of the data collection and interpretation of the data as that addressed under the Major and Minor points noted by the reviewer, and we have addressed each point below.
Comment #2. Also given the style and content of the manuscript this is rather a review-like manuscript, providing a good overview of current studies, but limited new findings. I therefor suggest to redact the manuscript into a review format and include also reports that do not unidirectionally support the outline hypothesis.
Answer #2. Although the reviewer states that the manuscript is a review-like manuscript, we do not agree with the statement that this meta-analysis study presents "limited new findings." In fact, this study presents an exhaustive compilation of data in support of a rather foundational discovery often requiring more or less extensive computational conversions. This study shows that there is not only similarity in the concentration of ATP among cells and tissues, but that this millimolar concentration range transcends prokaryotic, archaeotic, and eukaryotic domains This observation, derived from an extensive meta-analysis is profound, with the paper supporting [documenting] a new intracellular function for ATP. Further, the data and the interpretation transcends the fields of biology, chemistry, biochemistry, and even evolution. We believe redacting this manuscript to the format of a review would be inappropriate.
Regarding the scope of our literature search, our inclusion criteria incorporated these reports involving a living source cell/tissue/organ or whole organism and further included data on measurements made on actual functioning tissues, such as ex vivo beating working rat heart or crystalline lenses within their intact capsule. Our rationale for excluding other types of modified cellular preparations are presented in the Methods section. There are two reasons that can be presented for limiting the database:
(1) In the cases we recovered, there are no clear interpretations, and certainly no ATP concentration data, relating such modified cellular preparations to performance of the same cells in an intact tissue. In contrast, we have determined intracellular concentrations of ATP in intact living tissues by direct comparisons of 31P NMR spectral concentrations that permit such measurements to concentrations obtained through classical extraction methods. Such spectral comparisons show that 31P NMR spectroscopy on intact living tissue produces the same tissue ATP concentrations as the classical methods to which it is compared, and these concentrations are in the millimolar range—not the micromolar range (REF 4, REF 1, Uncited REF Kopp SJ, Glonek T: Ex vivo P 31 NMR of lens, cornea, heart, and brain. Mag Reson Imag 1985;3:359 376).
(2) The studies of Schwarzkopf et al (2013 REF 97, 2015 REF 44) provide one instance where the same source tissue was examined by the same group of researchers studying the same type preparation using the current luciferin-luciferase assay for ATP. The concentrations of ATP between the two studies differed by three orders-of-magnitude, indicating problems with quantification of ATP in tissues. Moreover, there are studies reporting ATP concentrations in the millimolar range using the luciferin-luciferase ATP assay (for example REFS 18,22,25,36,48,50, and numerous others as listed in the column labeled "Assay Type" in Table S1). Based on the above It was our conclusion that micromolar ATP concentrations were more than likely anomalous (especially considering the fact that they were a thousand-fold less concentrated) and thus were excluded from our reported compilation. Similarly, examining human erythrocytes as listed in Table S1, by classical PCA extractions in two different studies, demonstrated mM concentrations of ATP, whereas a study using the luciferin-luciferase reaction produced low (0.02) micromolar concentration of ATP in erythrocytes. To further address this point we have added a paragraph on study limitations at the end of the Discussion.
Major points:
Comment #3. Overall, the study is based on a literature search with heavy subsequent filtering of hits. This filtering does not seem to be appropriate in all cases and in some places seem to be biased.
Answer #3. Although, the reviewer did not provide specific locations in the text where biases were apparent, we presume that the reviewer is referring to the elimination of studies for selective reasons, for example, where extremely small ATP concentrations in the micromolar range were reported. Our meta-analysis study focuses on intact tissues where manipulation of the intracellular or tissue's cellular constituency is not an issue. Although the filtering used in this study may not at first seem to be appropriate and in some cases it might be construed as being biased, the analyses covered multiple different analytical procedures that are presented in the Supplementary Table S1 under the heading "Assay Type". In contrast, in those studies where the micromolar concentrations of ATP are presented (not in Table S1), they are not derived from intact tissues and further make use of a single analytical method involving the luciferin-luciferase reaction. Because there may be three orders of magnitude of difference in the ATP concentration by this method (as also explained in Answer #2 above), relative to all the other methods presented in Table S1, we concluded that the reports using this method may somehow be biased relative to the bulk of the data acquired by others (see Table S1). It is for this reason we decided that we could not include these in the data base until the issue of why the luciferin-luciferase assays reported are so extremely low relative to intact tissue measurements.
It should be noted that in the various methods (see column Assay Type in Table S1) that produce millimolar concentrations of ATP in tissues almost exclusively involve treatment of the tissue with strong acid. This is not a characteristic of the luciferin-luciferase assay, and it is our contention that use of the luciferin-luciferase reaction misses a pool of ATP that is released from the tissues when the tissues are subjected to strong acid. The whole question becomes, what precisely is the luciferin-luciferase reaction determining in the cases of low ATP concentrations. It is our contention that this issue of the integrity of the analytical method is a subject that is far beyond the scope of this paper.
As was discussed above, it should be noted that we did find assays of one source sample, where the ATP was measured using both classical and luciferin-luciferase assay. In these reports, the tissue was treated with strong acid and produced millimolar concentrations of ATP (REF 44); whereas, in the second case, the procedure yielded only micromolar concentrations of ATP using the same luciferin-luciferase reaction for the detection of ATP (REF 97). In this instance, it appears that the question of the extraction procedure involved is the source of the discrepancy in the data. This is especially significant when the ATP measurement was made by the same research group.
Comment #4. E.g. why do the authors exclude “measurements of ATP in the cellular cytoplasm”. Selectively analyzing such measurements would be much more appropriate to substantiate the hypothesis of ATP serving a role as hydrotrope, rather bulk measurements of cellular ATP which “average” ATP concentrations across multiple intracellular organelles with highly divergent ATP levels and hence depend on organelle volume/density etc..
Answer #4. The focus in our paper, in so far as is possible, involved intact source cells/tissues/organs or whole organisms for the simple reason that dissection procedures of any kind, chemical or mechanical, will alter the ATP levels in the samples. There is no easy way of knowing how extensive tissue disruption is for any given tissue or sampling procedures. Therefore, we made the effort to examine cells and tissues in as natural a state as possible. For example, we used cases involving whole beating working hearts (Kopp SJ, Glonek T, Erlanger M, Perry EF, Bárány M, Perry, Jr, HM. Cadmium and lead effects on myocardial function and metabolism. J Environ Pathol Toxicol 1980;4:205‑227) or large pieces of skeletal muscle or in the case of the lens organ, the whole organ (Glonek T, Kopp SJ. Ex vivo P‑31 NMR of lens, cornea, heart, and brain. Mag Reson Imag 1985;3:359‑376) encapsulated by a membrane. Our more than 40 years (Bárány M, Bárány K, Burt CT, Glonek T, Terrell C. Myers. Structural changes in myosin during contraction and the state of ATP in the intact frog muscle. J Supramol Struct 1975;3:125‑140.) experience in these kind of experiments, told us that extensive or even mild manipulation of tissues, either mechanical or chemical, reduces ATP levels. Further, we admit that we have no idea how extensive ATP destruction would be while trying to analyze ATP on a microscope slide as can be the case when using the luciferin-luciferase reaction. Such explanations present a level of complexity that extends far beyond the scope of this manuscript and, therefore, we elected not to use such data.
Our collective knowledge of chemistry permits an in depth understanding of the micromolar range of ATP found in the bulk of the studies cited regarding ATP measurements, and although we agree that taken collectively, this is not a valid reason for exclusion, in each of these studies in which we conducted a thorough review, taken individually there are valid reasons for exclusion. To further address this point we have added a paragraph on study limitations at the end of the Discussion.
In summary, exclusion of the "measurements of ATP in the cellular cytoplasm" was done to prevent the influence of dissection on cells and tissues. In organs, this can be illustrated in our paper on the crystalline lens organ (REF 1; Uncited Greiner, JV, Kopp SJ, Glonek T: Distribution phosphatic metabolites in the crystalline lens. Invest Ophthalmol Vis Sci 1985;26:537-544) where it can be established that the sum of the metabolites of lens parts (cortex, nucleus, and capsule) can be close to the values of ATP in the intact lens but is not equal to the intact lens. Thus, we have observed that when cells/tissues/organs are disassembled, there may be unexpected and even profound changes in the ATP concentration. It is more than likely that by selective analysis, such measurements would not only introduce bad data into the data set, but its influence on the hypothesis of ATP serving a role as a hydrotrope (Greiner, Glonek REF 6). Introduction of such variability might alter or even change introduction of our hypothesis of ATP serving a role as hydrotrope.
The variability in ATP measurements among parts of cells, e.g., cytoplasm, nucleus, nucleoli, or mitochondria coupled with variance in cell parts due to cell organelle or cytoplasmic sample preparation (dissection techniques and or centrifugation techniques), in addition to the sparse number of reports on parts of same cells of same species, makes it even more difficult to obtain reliable numbers. This becomes even more problematic when considering the intracellular variations of organelle size or cytoplasmic volumes.
Comment #5. The authors also exclude studies reporting tissue ATP concentrations in the micromolar range based on the rationale that these are not in line with the bulk of other captured data. This is not a valid reason for exclusion as long as the technical approach for measuring ATP concentrations in these studies is sound.
Answer #5. We disagree with the reviewer, that measuring ATP-concentrations in studies reporting micromolar concentrations of ATP is sound, especially when conducting an in-depth analysis of the exact methods used in each of the excluded studies. Unless the luciferin-luciferase method is performed on tissues subjected to a strong acid or a strong chelating reagent to properly extract ATP and a micromolar concentration is still found, the methodology is questionable. This subject, however, is far and beyond the scope of this study. In Table S1, we have tabulated all of the methods employed in determining tissue ATP levels, some of which have been employed repeatedly in laboratories around the world and are, therefore, extremely well validated. The ATP levels being so close, testify to the repeatability of the measurement techniques and the millimolar concentration.
Again, as mentioned above, In order to contain the scope of the paper, we elected, in so far as possible, to work with intact, living tissues, whole organs, or whole organisms, e.g., perfused beating working hearts, lenses within their encapsulating basement membrane capsules and whole microorganisms.
Again, as mentioned above, In order to measure ATP levels in cellular cytoplasm involves dissection of a cell. As such, an analysis of cellular cytoplasm involves disruptions of the cell membrane and cellular contents which surely will influence ATP levels, for example, when a tissue specimen preparation is spun down in a centrifuge tube to obtain a supernatant solution.
Regarding the intact lens, this is an intact organ that is >90% protein. As such, the lens model is an excellent avascular model for studying the interaction of ATP with protein. Because there is not much in the way of substance of any other enzymatic activity in this organ, any measurement would be of the ATP and the protein. Other reactions involving ATP would be very minor. This in fact, is based in part on our interest in studying the lens intact organ (REF 6), was the impetus to present this study.
Comment #6. Even more so as the authors include studies in which the quantitative analytical technologies for measuring ATP were “not stated in the source study”.
Answer #6. Table S1, under the heading of "Assay Type," provides the analytical procedure used to obtain the ATP levels for every tabulated entry presented. When we could not find the method of assay for the ATP stated in the paper even by searching previous methods references offered by the authors, which was a fairly common occurrence, these ATP values/papers were excluded from the compilation of reports. This is a question of rigor when compiling values derived from the literature. In each instance where an ATP value was reported and only a reference given to analytical method, we returned and examined the source paper to be certain that the method was provided and suitably calibrated. The source paper was then cited in Table S1. To further clarify this point, a sentence has been entered on page 4 para lns 4-6 "When the method of assay for ATP was not stated in a published report, even by searching previous method references, these ATP values were excluded from this compilation of reports."
Comment #7. I rather suggest the authors provide a full list of reported ATP concentrations (in non-cell line specimens) and then discuss reasons for different outcomes rather than excluding the majority of studies a priori.
Answer #7. This suggestion of the reviewer, again, has been addressed in the preceding discussions. Such an extensive compilation would expand the study far beyond the scope of the paper as has been addressed in the previous comments. In regard to providing a list of "non cellular specimens", we presume the reviewer is referring to cell cultures and, as was mentioned previously, these were excluded because there were too many factors involved in trying to determine the integrity of the assay.
The second point is the very low ATP concentration values obtained using tissue cultures and the luciferin-luciferase ATP assay. This would require a significant expansion of this already large study and does not fulfill the mission/purpose of this study.
Comment #8. The main concern is the rather unidirectional interpretation of the data to support the hydrotrope hypothesis. Although interesting, the fact that ATP concentrations are high across all examined specimens is not a proof for (an evolutionary) hydrotrope function of ATP. If the authors want to support and substantiate such claims, they need to compare ATP concentrations in specific intracellular compartments (e.g. cytoplasm) between healthy specimens and diseased specimens in which disease is caused by protein aggregation/misfolding.
Answer #8. The reviewer is correct, our hypothesis is not proof for an evolutionary hydrotrope function of ATP, and we did not state that it was proof, but it is strongly suggestive in the support of this hypothesis.
Comment #9. The statistical comparison of different specimens and Table 3 are not really informative and do not add to the paper.
Answer #9. This table was included to demonstrate that regardless of the function of a tissue, the ATP concentration among the tissues is essentially a constant value. This is an observation, which on the surface, is counterintuitive.
Minor points:
Comment #10. Table 1 lists functions of ATP which the authors claim require only micromolar concentrations of ATP. Such concentrations, and also references supporting these findings, should be included in the table. Otherwise, it’s a bit like a textbook list.
Answer #10. We thank the reviewer for their comment and have added the words "believed to be" after the word "that" and before the word "required". This is a common, though not yet proven, belief. Although all of the listed items are known to represent functions of ATP, we could not find them in any one textbook list where all these functions are presented. Therefore, we felt that assembling these functions on one concise table would be helpful in understanding the many and diverse functions of ATP (Table 1).
Comment #11. The simple summary, contains a few grammatical errors. E.g.:
Line 7: the comma after prokaryotes should be replaced by a full stop.
Answer #11. We thank the reviewer. In reference to the comma on line 7, the comma has been replaced by a period ".".
Comment #12. ine 8/9: “(average~4.4 millimolar). regardless of how it lived or its function.” Should be changed to “(average~4.4 millimolar), regardless of how they lived or their function.”
Answer #12. We thank the reviewer. The reviewer's suggestions have been entered into the manuscript on page 1 lines 8-9.
Comment #13. Some of the datasets in table S1 appear redundant. E.g. multiple data extracted from refs 41 or 67 are identical. Were these ATP concentrations clearly obtained from individual samples or were same samples mentioned several times in the respective references
Answer #13. These ATP concentrations as they were presented in the report were clearly obtained from intact and individual samples and are not the same samples measured several different times. We thank the reviewer for pointing this out and as such, to clarify this, we have stated this by adding this in the manuscript on page 13 para 1 at the end of the Table S1 legend to read "All ATP concentrations were obtained from individual discrete samples."
However, since the Table S1 is unable to be altered in the format the journal provided we intend for the words "All ATP concentrations were obtained from individual discrete samples." be entered at the end of the Legend. We must ask for the journals help in how to add these words to the end of the Legend.
Reviewer 3 Report
Authors aim to explain through a literature search why high concentration of ATP was found in tissues like cristalline lens while no high energy needing occurs in such tissue. They compared results of ATP concentrations in different tissues including cardiac, skeletal muscle, brain , liver, retina and microorganisms. They postulate that such ATP concentration could have an other function: to inhibit protein aggregation through tits amphiphilic properties.
The study is very interesting and novel, however I have some comments
While the introduction is well done, materials and methods lacks some data.
Authors analyzed the level of ATP concentration between 1946 and 2020.
The method to dose ATP has dramatically change during a such enlarge period.
I am not sure that the ATP values evaluated with the method used in 1946 was the same today and give the same values. This should appear in a limitation study paragraph.
ATP measurement in laboratory cell lines was excluded. Do you mean because they are transgenic cell line that may disturb ATP concentration?
Why did authors excluded measurements of ATP in organelles and cytoplasm is not lear for me. Please explain better.
Results:
Why no literature data on cristalline lens was given in methods? (Ref 6) suggest a such dosage that not appear in the listing of table 2?
Legend of table 2 lacks informations. ATP concentration is given in mM of what? Is it whole cell extracts or supernatant after centrifugation or whole scared tissues?. This sis important because we can compare only what is comparable.
The LC MS MS remains the "gold standard" for ATP measurement see Xiapong Fu et al Anal Chem 2019 91: 5881-5887.
The ATP concentration was 2 to 3 nano mol per g of tissue
corresponding to 2 to 3 micomlar /L of tissue extracts.
please comment
Discussion
The question may be summarized by why ATP is so elevated in "inerte tissue " like cristalline lens. The hypothesis of preventing protein aggregation by improving protein solubility is very interesting and highly plausible but not demonstrated here. Because it is a review of the literature of course the aim was not to demonstrate but to list elements.
Author Response
Authors aim to explain through a literature search why high concentration of ATP was found in tissues like cristalline lens while no high energy needing occurs in such tissue. They compared results of ATP concentrations in different tissues including cardiac, skeletal muscle, brain, liver, retina and microorganisms. They postulate that such ATP concentration could have another function: to inhibit protein aggregation through tits amphiphilic properties.
The study is very interesting and novel, however I have some comments
While the introduction is well done, materials and methods lacks some data.
Authors analyzed the level of ATP concentration between 1946 and 2020.
Comment #1. The method to dose ATP has dramatically change during a such enlarge period.
Answer #1. We do not understand this question. The data presented in Table S1 was obtained from living cells/tissues/organs or whole organisms without any ATP interventions; as can be learned from each reference, there was no dosing of the tissue specimens.
Comment #2. I am not sure that the ATP values evaluated with the method used in 1946 was the same today and give the same values. This should appear in a limitation study paragraph.
Answer #2. As discussed with answers to reviewer 2, the values obtained and presented in Table S1 show that the values obtained by a host of Methods spanning decades are the same despite the fact that they use different analytical procedures and were performed in a variety of laboratories. This is the point of Table S1; it demonstrates that careful methods carried out in competent laboratories even back as far as 1946 are remarkably similar. However, a sentence regarding this comment has been added to the manuscript at the end of paragraph 2 in the Results section page 4 lns 6-10 "Moreover, the consistency of the data, including the methods used in Table S1, demonstrates the similarity in the ATP concentrations, and that regardless of the laboratory conducting the experiments or the analytical methods employed, the concentration of ATP is a constant in living cells/tissues/organs and whole organisms."
In summary, we agree with the reviewer that the methods used to measure ATP concentration in earlier years are different than those used currently. Also, it must be remembered that there is danger in some of the more modern methods, for example the luciferin luciferase reaction discussed in a previous paragraph.
Since the reviewer has suggested we also write a paragraph on study limitations and entered further discussion on this point in the manuscript this has been added as the last paragraph of the Discussion as follows: see page 7 para 3 "This study has limitations. Since the meta-analysis of the literature citing measurement of the ATP concentrations spans the period from 1946 through 2020, the results presented were derived from methods of measuring the ATP concentration that evolved with more precise technologies. However, with evolution of these methods, it must be cautioned that later methods, for example, the luciferin-luciferase reaction, require consideration and calibration to demonstrate their accuracy. This is particularly so when more recent values obtained using such methods are lower by 3 orders-of-magnitude than those determined earlier. Such reports as described in the Methods were not included in this meta-analysis. Other limitations as explained in the Methods were the exclusion of reports on ATP concentrations utilizing modified cellular/tissue preparations: these include cell lines, to avoid the possible influence of cellular abnormalities; use of cytoplasm and various intracellular organelles to avoid the effects of both mechanical and chemical disturbances induced as a result of sample procurement; reports with data in the micromolar range with exceedingly low concentrations of ATP inconsistent with the bulk of the captured data in the millimolar range (this phenomenon was often associated with conflicting re-ports employing the luciferin-luciferase reaction for measurement of ATP demonstrating very low ATP concentrations in the same tissue and even the same species). Although these selective limitations may result in some degree of bias, it is our contention that such limitations are justified."
Comment #3. ATP measurement in laboratory cell lines was excluded. Do you mean because they are transgenic cell line that ATP concentration?
Answer #3. The point about exclusion of cell lines was addressed in the Methods section beginning in line 7. The answer to the question is yes, since cell lines may not be normal as well as the ATP concentrations in these cells. We have added further clarification in the suggested limitations paragraph at the end of the Discussion section.
Comment #4. Why did authors excluded measurements of ATP in organelles and cytoplasm is not lear for me. Please explain better.
Answer #4. As presented for reviewer 2, exclusion of the "measurements of ATP in the cellular cytoplasm" was done to prevent the influence of dissection on cells and tissues. In organs, this can be illustrated in our paper on the crystalline lens organ (REF 1; Uncited Greiner JV, Kopp SJ, Glonek T: Distribution of phosphatic metabolites in the crystalline lens. Invest Ophthalmol Vis Sci 1985;26:537-544) where it can be established that the sum of the metabolites of lens parts (cortex, nucleus, and capsule) can be close to the values of ATP in the intact lens but is not equal to the intact lens. Thus, we have observed that when cells/tissues/organs are disassembled, there may be unexpected and even profound changes in the ATP concentration. It is more than likely that by selective analysis, such measurements would not only introduce bad data into the data set, but its influence on the hypothesis of ATP serving a role as a hydrotrope (Greiner, Glonek REF 6). Introduction of such variability might alter or even change introduction of our hypothesis of ATP serving a role as hydrotrope.
The variability in ATP measurements among parts of cells, e.g., cytoplasm, nucleus, nucleoli, or mitochondria coupled with variance in cell parts due to cell organelle or cytoplasmic sample preparation (dissection techniques and or centrifugation techniques), in addition to the sparse number of reports on parts of same cells of same species, makes it even more difficult to obtain reliable numbers. This becomes even more difficult a problem when considering the intracellular variations of organelle size or cytoplasmic volumes.
Comment #5. Why no literature data on cristalline lens was given in methods? (Ref 6) suggest a such dosage that not appear in the listing of table 2?
Answer #5. The number of lens published papers that would permit calculation of the concentration of ATP are limited to two. This number of papers did not warrant an independent category as described in our Methods section requiring at least >3 as stated on page 3 para 2 ln 5. As such, the lens is presented under the grouping "Other". However, each value of the lens ATP concentration found is entered into Table S1. The request of the reviewer is presented in Table S1 as references 75 and 1.
Further, it is important to know that although we reported numerous lens studies as referenced in REF 6, our studies were presented in percent mole-fraction, and total protein was not measured which would have allowed computation of molarity. As a consequence, absolute concentration values could not be calculated for the present study. We did, however, present the comparison of 31P NMR data to ATP extractions by classical perchloric acid extraction methods in both lens (REF 1) and muscle (REF 4) (mM ). and found such studies to be equivalent.
As for dosage in Table 2, there was no dosing in these reports.
Comment #6. Legend of table 2 lacks informations. ATP concentration is given in mM of what? Is it whole cell extracts or supernatant after centrifugation or whole scared tissues?. This sis important because we can compare only what is comparable.
Answer #6. The values reported in the present study are in millimolar in whole cells/tissues/organs and whole organisms. As explained in the text, only studies reporting data that could be reduced to an ATP concentration are cited. Of course, there are many studies where ATP is determined, but where there is no reference to an absolute concentration or any method to obtain such a value by mathematical computation(s) such reported values were not cited.
We thank the reviewer for this comment and in order to clarify this point we have added the words "normal whole source" after the word "all" in the description of Table 2 on page 3 line 1..
Comment #7. The LC MS MS remains the "gold standard" for ATP measurement see Xiapong Fu et al Anal Chem 2019 91: 5881-5887.
Answer #7. The authors thank the reviewer for the reference by Fu et al regarding ATP and liver tissue. However, after review, we found no quantitative data; there was no reported measurement of ATP concentration. Also, it should be noted that this new technique may be yet another example of why the more traditional methods of measuring ATP in the liver yield values in the millimolar range (see Table S1), while this new procedure produces values that are a thousand-fold smaller. As we commented above, we observed this discrepancy with the luciferin-luciferase reaction. It appears that the small value yield may have nothing to do with the analytical method per se, but with the extraction methods, as the luciferin-luciferase reaction requires that the ATP be in an aqueous solution. In contrast, the 31P NMR ex vivo method used in our laboratory (REF 1 and Uncited REF Kopp SJ, Glonek T: Ex vivo P-31 NMR of lens, cornea, heart, and brain. Mag Reson Imag 1985;3:359 376) is conducted on a functioning living tissue without any chemical intervention.
Comment #8. The ATP concentration was 2 to 3 nano mol per g of tissue
corresponding to 2 to 3 micomlar /L of tissue extracts.
please comment
Answer #8. As mentioned above for the paper by Fu et al, (1919), the concentration of ATP in the extract is not related to the concentration in the intact tissue. To review again, in the reference cited by Fu et al, there is no concentration of ATP in whole source intact tissue presented, nor does the paper report any combination of numbers from which the mM can be calculated. This work presents another analytical method that could be used for determination of nucleotides in extracts, but for the foregoing reasons we did not incorporate it into our current study.
Discussion
Comment #9. The question may be summarized by why ATP is so elevated in "inerte tissue " like cristalline lens. The hypothesis of preventing protein aggregation by improving protein solubility is very interesting and highly plausible but not demonstrated here. Because it is a review of the literature of course the aim was not to demonstrate but to list elements.
Answer #9. We agree with the reviewer's comment, and at this time, we have only the hypothesis presented to rationalize why this high concentration of mM ATP exists in lens tissues.
Round 2
Reviewer 2 Report
The authors have addressed some of the concerns raised by the reviewers. However, I am still not convinced that the unidirectional interpretation of the data with the strong emphasis of the hydrotrope hypothesis is valid and does justice to the rather general title.
What the paper clearly provides is a summary(and statistical analysis) of ATP concentrations in tissues/organs/specimens. The interpretation, however, is speculative and, in the way it is presented, very suggestive. As the authors state themselves in their reply, there is no proof to link the high intracellular ATP concentrations to the recent report of ATP acting as hydrotope in cell/tissue extracts. Therefore alternative explanations/hypothesis for the conserved, high intracellular ATP concentrations need to be considered and discussed with similar vigour.
Author Response
Comment. The authors have addressed some of the concerns raised by the reviewers. However, I am still not convinced that the unidirectional interpretation of the data with the strong emphasis of the hydrotrope hypothesis is valid and does justice to the rather general title.
What the paper clearly provides is a summary(and statistical analysis) of ATP concentrations in tissues/organs/specimens. The interpretation, however, is speculative and, in the way it is presented, very suggestive. As the authors state themselves in their reply, there is no proof to link the high intracellular ATP concentrations to the recent report of ATP acting as hydrotrope in cell/tissue extracts. Therefore alternative explanations/hypothesis for the conserved, high intracellular ATP concentrations need to be considered and discussed with similar vigour.
Answer. The interpretation of the data, which exhibits strong emphasis on the hydrotrope hypothesis, is elaborated upon in detail, in our paper on lens (Greiner and Glonek: Hydrotropic function of ATP in the crystalline lens. Exp Eye Res 2020;190:107862) which is reference #6 in the manuscript. Further, the report by Patel et al. (ATP as a biological hydrotrope. Science 2017;356:753-756), which is reference 5 in our manuscript, provides multiple and extensive studies pertaining to the hydrotrope hypothesis. Our ex vivo studies in the living crystalline lens organ (Glonek, Kopp, Greiner: Phospohrus-31 NMR of the intact crystalline lens.: 1. The living lens spectrum. II. The spectroscopic effects of deuterium oxidize incubation. Phosphorus Sulfur 1983;18:329-332; Glonek and Greiner: Intralenticular water interactions with phosphate in the intact crystalline lens. Ophthalmic Res 1990;22:302-309) provides even further supporting physicochemical evidence for ATP as a hydrotrope. (See reference #6 for further explanations.) Considering, however, our study's general title along with the above referenced studies, upon considering the reviewers comment, we believe that the plural term "implications" should be changed to the singular "implication" so as not to imply that we are reporting more than one implication.
Although our interpretation of the data in the present study might be considered speculative, careful judgment and evaluation of the multiple experiments documented in detail in the supplementary materials in the paper by Patel et al. (2017), coupled with the experiments by our group in support of hydrotropic ATP explained in detail in our paper (reference #6), we have established the key relationships between the experiments. Reviewing what already has been presented, it is difficult to formulate an alternative explanation as to what other hypothesis might be developed regarding our finding of such a high concentrations of ATP. We agree there is no proof to directly link the high intracellular ATP concentrations to the recent reports of ATP acting as a hydrotrope in cell/tissues (references 5 and 6). To our knowledge, no such reports exist. We agree with the reviewer that alternative explanations and hypotheses for the conserved high intracellular ATP concentrations should be considered and discussed; however, we are unaware of any alternative explanations or hypotheses for these high concentrations of ATP. This inability to find any supportable hypothesis is significant, since both Dr. Glonek and I have spent collectively over 90 years in researching ATP. In fact, Dr. Glonek is considered a leading authority on ATP, since his original thesis work beginning in 1967 is on ATP. Moreover, when reviewing our paper in Exp Eye Res (reference #6) it specifically spells out the difficulty in attempting to explain the etiology of this millimolar concentration. This also is presented in the Introduction section of our paper. It is clearly developed in the Introduction of the paper that there are no explanations available until the work of Patel et al., 2017 in Science. The interpretation may be unidirectional; however, to our knowledge, there are no other remotely plausible explanations or hypotheses available, and as such no discussion, as suggested, is possible at this time.
As to whether or not ATP is a hydrotrope, we refer to the work by Patel et al. (2017) in numerous experiments coupled with the elaborate explanations we provide (reference #6), including two very convincing experiments of our own that support Patel et al, using actual intact living lens tissue. The lens is unique for such hydrotropic studies since the crystalline lens is primarily comprised of protein. The lens organ relies on a strict ordering of proteins in order to maintain a state of non-aggregation of proteins, and this ordering is required for lens functional transparency. It has been clearly shown that there is a relationship between maintaining proteins in the non-aggregated state in order to avoid cataract formation. Protein aggregation is more than likely the etiology of cataractogenesis in the aging lens. This concept is reviewed and referenced in reference #6. More importantly, it must be understood that ATP is chemically a hydrotropic molecule, including all the essential atomic and structural features of a hydrotrope [Neuberg 1916]. So, the issue is not just, is a molecule of ATP a hydrotrope, but does ATP bind to proteins and help solubilize them as demonstrated in multiple experiments and most important the supplementary materials in the Patel et al. (2017) paper in Science.
As stated in our paper in Exp Eye Res (reference #6), the phenomenon of the high unexplained concentrations of ATP in intact organs has perplexed us. We have searched for an explanation for over 40 years ever since our discovery of the high concentration of ATP in the living crystalline lens using phosphorus-31 NMR (Greiner, Kopp, Sanders, Glonek: Organophosphates of the crystalline lens. A nuclear magnetic resonance spectroscopy study. Invest Ophthalmol Vis Sci 1981;21:700-713), more recently this discovery of the similarity of high mM concentrations of ATP among different tissues resulted in our, again, scanning the literature as we have done for the ATP concentrations presented in the manuscript. Our literature searches revealed that there are no other advance theories to explain these high ATP concentrations. Since we have no de novo explanations or know of any other explanations, it is impossible to do what the reviewer requests.
This being said, however, in consideration of the comment made by the reviewer, we have added a sentence regarding this point to the Discussion section at the end of the paragraph on limitations of our study which reads "Although our interpretation of the data presented herein is albeit unidirectional, the basis that ATP is acting as a hydrotrope preventing protein aggregation in living systems is established by multiple experiments [references 5 and 6]. In the absence of any other explanation or hypotheses, we have presented our hypothesis for the function of high concentrations of ATP in the intact crystalline lens and other cells/tissues."
Respectfully submitted,
Jack V. Greiner, OD, DO, PhD
Thomas Glonek, PhD